# Adaptive Cross Batch Normalization for Metric Learning

## Abstract

Metric learning is a fundamental problem in computer vision that aims to learn a semantically useful embedding space via ranking losses. Traditionally, the effectiveness of a ranking loss depends on the minibatch size, and therefore, it is inherently limited by the memory constraints of the underlying hardware. While simply accumulating the embeddings across minibatches proved useful (Wang et al. (2020)), we show that it is equally important to ensure that the accumulated embeddings are up to date. In particular, it is necessary to circumvent the *representational drift* between the accumulated embeddings and the feature embeddings at the current training iteration as the learnable parameters are being updated. In this paper, we model this representational drift as a transformation of the distribution of feature embeddings and approximate it using the first and second moments of the empirical distributions. Specifically, we introduce a simple approach to adapt the stored embeddings to match the first and second moments of the current embeddings at each training iteration. Experiments on three popular image retrieval datasets, namely, SOP, In-shop and DeepFashion2, demonstrate that our approach significantly improves the performance on all scenarios.

## 1 Introduction

Metric learning is a fundamental problem in computer vision with various applications including image retrieval (Yang et al. (2017); He et al. (2018)), zero shot learning (Bucher et al. (2016)), face recognition (Wen et al. (2016)), and visual tracking (Tao et al. (2016)). The main objective is to learn a metric space where the feature embeddings of instances from the same class are closer than the ones from different classes.

Since metric learning requires comparing the feature embeddings of different instances, the loss functions used for this problem (Chopra et al. (2005); Schroff et al. (2015); Brown et al. (2020); Kim et al. (2020)) are commonly referred to as ranking losses. Specifically, a ranking loss is computed by comparing the embedding of each instance in the minibatch against all the embeddings of a reference set (usually the minibatch itself). To improve the effectiveness of learning, the most informative samples are selected from the reference set via sophisticated mining strategies (Harwood et al. (2017); Suh et al. (2019)).

Note that, the larger the reference set, the more accurate the ranking loss computation. However, the reference set size is limited by the memory and computational constraints of the underlying hardware. Therefore, we require a scalable approach to use a larger reference set (preferably as large as the training set) while ensuring that the embeddings of the reference set are up to date. Cross Batch Memory (XBM) (Wang et al. (2020)) is a recent technique to expand the reference set by accumulating embeddings across minibatches. While this approach is efficient, *it does not ensure that the embeddings are up to date.* When the model is being trained (especially in the beginning of training) the embeddings evolve rapidly and the previous embeddings quickly become outdated. Such outdated embeddings not only limits the full potential of using a larger reference set but also provide contradicting supervision signal which may lead to suboptimal learning.

In this paper, we introduce a technique to tackle the *representational drift* in the XBM approach to effectively make use of a larger reference set. Specifically, we adopt a Bayesian framework and model the representational drift as a transformation of the distribution of the embeddings. To this end, the ideal transformation to

mitigate representational drift is to ensure that the embeddings of the reference set (*i.e.*, cross batch memory) follow the same distribution as the embeddings of the full dataset at any given iteration. For practical purposes we represent the empirical distributions of embeddings using their respective first and second moments (*i.e.*, mean and standard deviation).

We first introduce Cross Batch Normalization (XBN), which simply adapts the embeddings in the reference set to match the first and second moments of the current minibatch embeddings at each training iteration. Then, since the minibatch statistics can be noisy, to better estimate the dataset statistics at any given iteration, we adopt a Kalman filter approach (Kalman (1960)). Specifically, we use Kalman filter to estimate the mean and standard deviation of the dataset embeddings using the minibatch observations and the estimated statistics are used to adapt the reference set embeddings. We refer to this approach as Adaptive Cross Batch Normalization (AXBN).

We provide extensive experiments on three popular image retrieval datasets, namely, SOP (Oh Song et al. (2016)), IN-SHOP (Liu et al. (2016)) and DEEPFASHION2 (Ge et al. (2019)). Our results demonstrate significant improvements over the XBM method in all scenarios, which confirms our hypothesis and the merit of our approach. Furthermore, while the simpler XBN approach outperforms traditional XBM, the adaptive version is better in the small batch regime, albeit by a small margin, providing evidence that classical noise modelling approaches can be useful in deep learning.

## 2 Preliminaries

In this section, we discuss metric learning in the context of image retrieval, although it has a wide variety of applications. Let $\mathcal{D} = \{(\mathbf{x}_i, y_i)\}_{i=1}^n$ be the dataset where $\mathbf{x}_i \in \mathcal{X}$ is the input image corresponding to sample $i$ and $y_i \in \mathcal{Y}$ is the label. The objective of metric learning is to learn an embedding function $f_{\mathbf{w}} : \mathcal{X} \to \mathbb{R}^d$ by optimizing the parameters $\mathbf{w} \in \mathbb{R}^m$, such that images corresponding to the same label are clustered together in the $d$-dimensional embedding space. For notational convenience we write $\mathbf{z} := f_{\mathbf{w}}(\mathbf{x}) \in \mathbb{R}^d$. Ideally, for any triplet $(a, p, n)$ with $y_a = y_p \neq y_n$ we want $\|\mathbf{z}_a - \mathbf{z}_p\| < \|\mathbf{z}_a - \mathbf{z}_n\|$ for some appropriate distance metric $\| \cdot \|$. Once trained, given a set of gallery images, their corresponding embeddings are computed and stored[1]. During testing, a new query image is passed through the embedding function and its k-nearest neighbours in the embedding space are retrieved from the stored gallery embeddings.

The problem of metric learning can be written as:

$$\text{minimize}_{\mathbf{w}} \, L(\mathbf{w}; \mathcal{D}) := \frac{1}{n} \sum_{i=1}^n \ell(\mathbf{w}, (\mathbf{x}_i, y_i); \mathcal{D}) \, , \tag{1}$$

where $\ell$ denotes the embedding function composed with a metric learning loss. Note that in addition to $\mathbf{w}$ and the sample $(\mathbf{x}_i, y_i)$, $\ell$ also depends on the full dataset $\mathcal{D}$. Specifically, metric learning requires comparing the embedding of a sample with all other embeddings to ensure that the samples belong to the same class are clustered together while other class samples are pushed away. For example, the commonly used triplet loss (Schroff et al. (2015)) computes

$$\ell^{\text{triplet}}(\mathbf{w}, (\mathbf{x}_i, y_i); \mathcal{D}) := \frac{1}{|\mathcal{T}_i|} \sum_{(i,p,n) \in \mathcal{T}_i} \left[ \|\mathbf{z}_i - \mathbf{z}_p\|_2 - \|\mathbf{z}_i - \mathbf{z}_n\|_2 + \alpha \right]_+ \, , \tag{2}$$

where $\mathcal{T}_i$ denotes the set of all triplets for sample $i$ such that $y_i = y_p \neq y_n$, $\alpha \geq 0$ is a margin, $\| \cdot \|_2$ is the $L_2$ norm, and $[\cdot]_+$ projects onto the nonnegative real numbers.

Since the dataset is large, it is infeasible to compare all the samples in a dataset at each iteration. To this end, the standard approach is to use a reference set $\mathcal{R}^k \subset \mathcal{D}$ in place of the whole dataset for each optimization iteration $k$. Additionally, minibatch based Stochastic Gradient Descent (SGD) is employed to optimize the

---

[1]The set of gallery embeddings are sometimes referred to as the index.

loss function. Putting these two together, the SGD update equation at iteration $k$ can be written as:

$$\mathbf{w}^{k+1} = \mathbf{w}^k - \eta^k \, \nabla L \left( \mathbf{w}^k; \mathcal{B}^k, \mathcal{R}^k \right) \;, \tag{3}$$

$$L \left( \mathbf{w}^k; \mathcal{B}^k, \mathcal{R}^k \right) \coloneqq \frac{1}{|\mathcal{B}^k|} \sum_{(\mathbf{x}, y) \sim \mathcal{B}^k} \ell \left( \mathbf{w}^k, (\mathbf{x}, y); \mathcal{R}^k \right) \;. \tag{4}$$

Here, $\eta^k > 0$ is the learning rate, $\mathcal{B}^k$ is the minibatch, and $\mathcal{R}^k$ is the reference set. In almost all metric learning scenarios the minibatch itself is used as the reference set, *i.e.*, $\mathcal{R}^k = \mathcal{B}^k$.

## 2.1 Cross Batch Memory (XBM)

The idea of XBM (Wang et al. (2020)) is to expand the reference set beyond the minibatch. To alleviate the computational complexity of performing the forward pass on a larger reference set, the insight is to store the embeddings computed during the previous iterations and use them as the reference set.

Let $\bar{\mathcal{B}}^k = \{ (\mathbf{z}^k \coloneqq f_{\mathbf{w}^k}(\mathbf{x}), y) \mid (\mathbf{x}, y) \in \mathcal{B}^k \} \subset \mathbb{R}^d \times \mathcal{Y}$ denote the embeddings and their labels corresponding to the minibatch $\mathcal{B}^k$ at iteration $k$. Then the analogous reference set of embeddings $\bar{\mathcal{R}}^k$ is the union of the most recent $M$ embeddings[2], where $|\bar{\mathcal{R}}^k| \leq M$ is the limit on reference set size. In certain cases, $M$ can be as large as the dataset itself. These accumulated embeddings $\bar{\mathcal{R}}^k$ are used to optimize the loss function, specifically the XBM loss function at each iteration $k$ can be written as $L \left( \mathbf{w}^k; \mathcal{B}^k, \bar{\mathcal{R}}^k \right)$.

The benefit of XBM relies heavily on the assumption that the embeddings evolve slowly. In the XBM paper, it is empirically shown that the features evolve slowly after a certain number of training iterations (refer to Fig. 3 in Wang et al. (2020)), where feature drift at iteration $k$ is defined as:

$$D(\mathbf{x}, k, \Delta k) \coloneqq \left\| \mathbf{z}^k - \mathbf{z}^{k - \Delta k} \right\|_2 \;, \tag{5}$$

where $\mathbf{z}^k$ is the embedding vector for the image $\mathbf{x}$, $\Delta k$ is the iteration step, and $\| \cdot \|_2$ is the $L_2$ norm. We note that this quantity is merely an empirical diagnostic and is not practical to compute during training to determine whether the features have become outdated or not. Furthermore, even though the features may change slowly in absolute terms, the relative order of elements (*i.e.*, rank) in the reference set can change dramatically. Moreover, the drift accumulates over time (large $\Delta k$), which is problematic when the minibatch size is small, relative to the size of the cross batch memory. Altogether this could lead to an inaccurate supervision signal from the ranking loss.

As noted in Wang et al. (2020), the *slow-drift* assumption is violated in the early phase of training. While we believe the initial training phase is most important, as discussed above, the embeddings in the reference set $\bar{\mathcal{R}}^k$ can become outdated even in the slow-drift regime. Such outdated embeddings not only limits the full potential of using a larger reference set, but also provides contradicting supervision signal degrading performance.

Therefore, we believe, it is important to ensure that the embeddings in the reference set are up to date throughout the whole training process.

## 3 Adapting the Cross Batch Memory

Our idea is to adapt the embeddings in the cross batch memory at each iteration to circumvent the representational drift between the embeddings in the reference set $\bar{\mathcal{R}}^k$ and the current minibatch $\bar{\mathcal{B}}^k$. Here the term *representational drift* refers to the notion that statistics computed on batches of embeddings vary as training progresses since $f_{\mathbf{w}^k}(\mathbf{x}) \neq f_{\mathbf{w}^{k'}}(\mathbf{x})$ where $\mathbf{w}^{k'} \neq \mathbf{w}^k$ are model parameters from some previous iteration $k' < k$. We model the representational drift as a (linear) transformation of the distribution of the embeddings. Then, the ideal transformation to mitigate representational drift is to ensure that the embeddings of the reference set (*i.e.*, cross batch memory) follow the same distribution as the embeddings of the full dataset at any given iteration.

---

[2]For brevity we refer to $\bar{\mathcal{R}}^k$ as the reference set embeddings, however it contains the embeddings and the corresponding labels.

For practical purposes, we represent the empirical distributions of embeddings with their respective first and second moments (*i.e.*, mean and standard deviation). To this end, at any iteration, we intend to ensure that the mean and standard deviation of the embeddings of the reference set match the mean and standard deviation of the embeddings of the dataset at that iteration.

## 3.1 Cross Batch Normalization (XBN)

In this section, we outline our algorithm with a simplifying assumption that the minibatch statistics match the statistics of the dataset. In the subsequent section, we discuss an approach to circumvent this assumption.

Since we only have access to a minibatch of data at each iteration, we simply adapt the embeddings of the reference set to have the mean and standard deviation of the embeddings of the current minibatch. Suppose $\mathbb{E}[\cdot]$ and $\sigma[\cdot]$ denote the mean and standard deviation of the embeddings, then Cross Batch Normalization (XBN) can be written as:

$$\hat{\mathbf{z}}^k = \frac{\mathbf{z}^k - \mathbb{E}[\bar{\mathcal{R}}^k]}{\sigma[\bar{\mathcal{R}}^k]} \sigma[\bar{\mathcal{B}}^k] + \mathbb{E}[\bar{\mathcal{B}}^k] \ , \qquad \text{for each } (\mathbf{z}^k, y) \in \bar{\mathcal{R}}^k \ . \tag{6}$$

Here, the division and multiplication are performed elementwise. After the adaptation, the current batch embeddings are added to the cross batch memory and the combined set is used as the reference set to compute the metric learning loss. This updated reference set is stored in memory and used in the subsequent iteration, ensuring that at every iteration, the cross batch memory is only an iteration behind the current batch.

Note that, this is just one additional line to the XBM code, but as can be seen in the experiments, it improves the results significantly.

### 3.1.1 Justification for XBN

Let $\mathbf{z} = f_{\mathbf{w}}(\mathbf{x})$ and $\mathbf{z}' = f_{\mathbf{w}'}(\mathbf{x})$ be two embeddings for the same image $\mathbf{x}$ computed using different parameters of the model, $\mathbf{w}$ and $\mathbf{w}'$, respectively. Assume that $\mathbf{z}' = g(\mathbf{z})$ for some unknown function $g := f_{\mathbf{w}'} \circ f_{\mathbf{w}}^{-1}$.[3] We can approximate $\mathbf{z}'$ by the first-order Taylor expansion of $g$ around an arbitrary point $\mathbf{z}_0$ as follows:

$$\mathbf{z}' \approx g(\mathbf{z}_0) + \nabla g(\mathbf{z}_0)^T (\mathbf{z} - \mathbf{z}_0) \ , \tag{7}$$
$$= \mathbf{A}\mathbf{z} + \mathbf{b} \ .$$

To estimate coefficients $\mathbf{A}$ and $\mathbf{b}$ we need quantities that can be estimated independently from either the minibatch or the cross batch memory. By the method of moments (Casella & Berger (2021)),

$$\boldsymbol{\mu}' = \mathbb{E}[\mathbf{z}'] = \mathbb{E}[\mathbf{A}\mathbf{z} + \mathbf{b}] = \mathbf{A}\boldsymbol{\mu} + \mathbf{b} \ , \tag{8}$$
$$\boldsymbol{\Sigma}' = \mathbb{E}[(\mathbf{z}' - \boldsymbol{\mu}')(\mathbf{z}' - \boldsymbol{\mu}')^T] = \mathbb{E}[\mathbf{A}(\mathbf{z} - \boldsymbol{\mu})(\mathbf{z} - \boldsymbol{\mu})^T \mathbf{A}^T] = \mathbf{A}\boldsymbol{\Sigma}\mathbf{A}^T \ . \tag{9}$$

Here, $\boldsymbol{\mu}$ and $\boldsymbol{\Sigma}$ denote the mean and the covariance and the expectations are taken over samples drawn from the same distribution. However, in practice, we estimate $\boldsymbol{\mu}$ and $\boldsymbol{\Sigma}$ from the reference set $\bar{\mathcal{R}}^k$, and $\boldsymbol{\mu}'$ and $\boldsymbol{\Sigma}'$ from the minibatch $\bar{\mathcal{B}}^k$, at each iteration $k$. Solving for $\mathbf{A}$ and $\mathbf{b}$ we have,

$$\mathbf{A} = \left(\boldsymbol{\Sigma}'\boldsymbol{\Sigma}^{-1}\right)^{\frac{1}{2}} \ , \tag{10}$$
$$\mathbf{b} = \boldsymbol{\mu}' - \left(\boldsymbol{\Sigma}'\boldsymbol{\Sigma}^{-1}\right)^{\frac{1}{2}} \boldsymbol{\mu} \ . \tag{11}$$

Assuming independence in components of $\mathbf{z}$ we can consider scalar equations:

$$z'_j = A_{jj} z_j + b_j = \frac{\sigma'_j}{\sigma_j} z_j + \mu'_j - \frac{\sigma'_j}{\sigma_j}\mu_j = \sigma'_j \frac{z_j - \mu_j}{\sigma_j} + \mu'_j \ , \tag{12}$$

---

[3]The function $g$ may not actually exist if $f$ is not injective, which is typically the case. Nevertheless, we can still estimate the approximation to a notional $g$.

where $\sigma_j := \Sigma_{jj}^{\frac{1}{2}}$ is the standard deviation. This is same as our adaptation formula (Eq. (6)). Here, we are taking a linear approximation of the transformation function $g$. Nevertheless, higher-order Taylor series expansions, requiring higher-order moments to be calculated, may give better results with increased complexity, but we do not consider that direction further in this paper.

## 3.2 Adaptive Cross Batch Normalization (AXBN)

Note that, we intend to circumvent the representational drift by adapting the statistics of the reference set to match the *statistics of the dataset* at each iteration. Since we only have access to a minibatch of data at each iteration, we only have a noisy estimation of the dataset statistics. Therefore, we adopt a Bayesian framework to model the process of estimating dataset statistics from minibatches. With the dataset statistics in hand, we can simply transform the reference set embeddings to match the estimated dataset statistics.

### 3.2.1 Kalman Filter based Estimation of Dataset Statistics

The Kalman filter is a special case of recursive Bayesian filter where the probability distributions are assumed to be Gaussians (Kalman (1960); Roweis & Ghahramani (1999)). To this end, we first briefly review Bayesian filtering and then turn to the Kalman filter.

**Recursive Bayesian Filter.** Let $\mathbf{u}$ be the random variable correspond to a dataset statistic (*e.g.*, mean or standard deviation of the embeddings) that we want to estimate and $\mathbf{v}$ be that statistic computed using the minibatch of data. Bayesian filtering assumes the Markovian property where the probability of the current state given the previous state is conditionally independent of the other earlier states (Chen et al. (2003)). With this assumption, let $p(\mathbf{u}_k \mid \mathbf{u}_{0:k-1}) = p(\mathbf{u}_k \mid \mathbf{u}_{k-1})$ be the process noise distribution and $p(\mathbf{v}_k \mid \mathbf{u}_k)$ be the measurement noise distribution at iteration $k$. Since $\mathbf{u}$ is hidden and we only get to observe $\mathbf{v}$, one can estimate the dataset statistic at iteration $k$ given the past observations as:

$$p(\mathbf{u}_k \mid \mathbf{v}_{1:k-1}) = \int p(\mathbf{u}_k \mid \mathbf{u}_{k-1}) p(\mathbf{u}_{k-1} \mid \mathbf{v}_{1:k-1}) d\mathbf{u}_{k-1} \ . \tag{13}$$

Now, with the current observation $\mathbf{v}_k$ the updated estimation becomes:

$$p(\mathbf{u}_k \mid \mathbf{v}_{1:k}) = \frac{p(\mathbf{v}_k \mid \mathbf{u}_k) p(\mathbf{u}_k \mid \mathbf{v}_{1:k-1})}{p(\mathbf{v}_k \mid \mathbf{v}_{1:k-1})} \ . \tag{14}$$

Once $p(\mathbf{u}_k \mid \mathbf{v}_{1:k})$ is estimated, the mean of the distribution can be taken as the estimated value. This is a general form of Bayesian filtering and depending on the application, certain assumptions have to be made (*e.g.*, the type of probability distributions, their initial values, *etc.*)

**Kalman Filter.** As noted earlier, the Kalman filter (Kalman (1960)) assumes the distributions are Gaussians and for simplicity we assume a static state model. Let us consider the case where we estimate the mean of the embeddings of the dataset at iteration $k$. Then $\mathbf{u}_k, \mathbf{v}_k \in \mathbb{R}^d$. Let $\hat{\mathbf{u}}_k$ be the estimate of $\mathbf{u}_k$ and $\mathbf{Q}, \mathbf{R}, \mathbf{P} \in \mathbb{R}^{d \times d}$ be the process covariance, the measurement covariance and the estimation covariance, respectively. Now, the distributions used in Eq. (14) take the following forms:

$$p(\mathbf{u}_k \mid \mathbf{u}_{k-1}) = \mathcal{N}(\mathbf{u}_{k-1}, \mathbf{Q}_k) \ , \tag{15}$$

$$p(\mathbf{v}_k \mid \mathbf{u}_k) = \mathcal{N}(\mathbf{u}_k, \mathbf{R}_k) \ , \tag{16}$$

$$p(\mathbf{u}_{k-1} \mid \mathbf{v}_{1:k-1}) = \mathcal{N}(\hat{\mathbf{u}}_{k-1}, \mathbf{P}_{k-1}) \ , \tag{17}$$

where $\mathcal{N}(\mu, \Sigma)$ denotes the multi-variate Gaussian distribution. The Kalman filtering steps are then:

$$\hat{\mathbf{u}}_{k,k-1} = \hat{\mathbf{u}}_{k-1} \ , \qquad\qquad \text{predicted state estimate} \tag{18}$$

$$\mathbf{P}_{k,k-1} = \mathbf{P}_{k-1} + \mathbf{Q}_k \ , \qquad\qquad \text{predicted variance estimate} \tag{19}$$

$$\mathbf{K}_k = \mathbf{P}_{k,k-1}(\mathbf{P}_{k,k-1} + \mathbf{R}_k)^{-1} \ , \qquad\qquad \text{Kalman gain} \tag{20}$$

$$\hat{\mathbf{u}}_k = \hat{\mathbf{u}}_{k,k-1} + \mathbf{K}_k(\mathbf{v}_k - \hat{\mathbf{u}}_{k,k-1}) \ , \qquad\qquad \text{updated state estimate} \tag{21}$$

$$\mathbf{P}_k = (\mathbf{I} - \mathbf{K}_k)\mathbf{P}_{k,k-1} \ . \qquad\qquad \text{updated variance estimate} \tag{22}$$

Here, $\hat{\mathbf{u}}_{k,k-1}$ and $\mathbf{P}_{k,k-1}$ denote the intermediate state and noise variance estimates. In our approach, we assume that the dimensions are independent so that $\mathbf{Q}$, $\mathbf{R}$, and $\mathbf{P}$ are diagonal matrices (and each dimension can be processed independently), which has computational advantages for high dimensional spaces.

For our estimation model, we need to initialize the estimation variance $\mathbf{P}$ and most importantly the process noise $\mathbf{Q}$ and measurement noise $\mathbf{R}$ needs to be identified. Even though there are some heuristic approaches to estimate these noise covariances (Odelson et al. (2006)), for simplicity, we treat them as constant hyperparameters in our implementation and use the same hyperparameter value for all dimensions to limit the number of hyperparameters. In summary, there are three hyperparameters $p_0, q$, and $r$, where $\mathbf{P}_0 = p_o\,\mathbf{I}, \mathbf{Q}_k = q\,\mathbf{I}$ and $\mathbf{R}_k = r\,\mathbf{I}$.

It is known that the initial estimation variance $p_0$ is not important as the Kalman filter converges quickly. Moreover, for scalar systems, it can be shown that the static system depends on $\lambda = r/q$ (Formentin & Bittanti (2014)). Therefore, essentially there is only one hyperparameter to be tuned. In our case, measurement noise is inversely proportional to the minibatch size. Therefore, we use $r/|\mathcal{B}^k|$ as the measurement noise, tune $r$ only, and the same value can be used with different minibatch sizes. Note that, when the measurement noise is assumed to be zero ($r = 0$), this process specializes to directly using the minibatch statistics (refer to Section 3.1). Our final update in its simplest form is provided in Appendix E.

### 3.2.2 Adaptation

Suppose $\hat{\boldsymbol{\mu}}_k$ and $\hat{\boldsymbol{\sigma}}_k$ be the estimated mean and variance of the dataset at iteration $k$ from the Kalman filter. Then the adaptation can be written as:

$$\hat{\mathbf{z}}^k = \frac{\mathbf{z}^k - \mathbb{E}\big[\bar{\mathcal{R}}^k\big]}{\sigma\big[\bar{\mathcal{R}}^k\big]}\hat{\boldsymbol{\sigma}}_k + \hat{\boldsymbol{\mu}}_k \;, \qquad \text{for each } (\mathbf{z}^k, y) \in \bar{\mathcal{R}}^k \;, \tag{23}$$

where $\mathbb{E}\big[\bar{\mathcal{R}}^k\big]$ and $\sigma\big[\bar{\mathcal{R}}^k\big]$ denote the mean and standard deviation of embeddings in the reference set. After the adaptation, the current batch embeddings are added to the cross batch memory and the combined set is used as the reference set to compute the metric learning loss.

We would like to emphasize that our approach does not add any overhead to the memory and computational requirements of the XBM method. Furthermore, our adaptation is a simple change to the XBM code which results in significant performance improvements.

## 4 Related Work

**Metric Learning Methods.** Recent advances in metric learning are mostly on improving the effectiveness of learning by modifying the loss function and/or the example mining strategy. Based on how embeddings of different instances are compared against each other, the loss functions can be broadly categorized into 1) pair/triplet based approaches (Chopra et al. (2005); Schroff et al. (2015); Sohn (2016); Wang et al. (2019); Khosla et al. (2020)), 2) methods that directly optimize average precision (Cakir et al. (2019); Brown et al. (2020)), and 3) proxy-based losses (Movshovitz-Attias et al. (2017); Qian et al. (2019); Kim et al. (2020)). It is worth noting that the differences in these loss functions impact the learning effectiveness, however, they are theoretically equivalent.

Apart from the loss, example mining strategy is important to ensure that the deep learning model focuses more on the informative examples. In addition to the popular (semi-)hard mining, there are many sophisticated strategies have been developed recently (Hermans et al. (2017); Harwood et al. (2017); Suh et al. (2019); Wang et al. (2019)). We only mentioned a few works here and we refer the interested reader to (Musgrave et al. (2020a;b)) for a comprehensive list of metric learning losses and example mining strategies.

In contrast to these works, we focus on expanding the reference set which allows us to compare more examples across minibatches while benefiting from these recent advances.

**Using External Memory.** Using external memory in metric learning is not new (Vinyals et al. (2016); Li et al. (2019); Zhong et al. (2019); Wang et al. (2020)). However, an important distinction is that we use an

external memory to expand the reference set and ensure that those embeddings are kept up to date so that they can be used for effective learning. Similarly, in self-supervised learning, Momentum Contrast (MoCo) (He et al. (2020)) uses an external memory along with a separate encoding network to compute the features and uses momentum based updates to ensure slow-drift of encoder parameters. Another conceptually similar approach is cross iteration batch normalization (Yao et al. (2021); Ioffe & Szegedy (2015)), which estimates minibatch statistics across training iterations using first order information. In contrast to both of these methods, we simply store the feature embeddings at each iteration and directly tackle the representational drift by adapting the distribution parameters for effective metric learning.

## 5 Experiments

We first describe the experimental setup and the datasets and then discuss the results.

### 5.1 Experimental Setup

We follow the standard approach and use an ImageNet (Russakovsky et al. (2015)) pretrained ResNet-50 (He et al. (2016)) backbone and set the embedding dimension $d$ to 512. ResNet-50 architecture is used as is and therefore batch normalization (Ioffe & Szegedy (2015)) layers are enabled. We implemented our code in the PyTorch framework (Paszke et al. (2019)) and made use of the Pytorch Image Models (TIMM) (Wightman (2019)) library for training pipline including data augmentations and pretrained weights. For metric learning specific components including the implementation of XBM, we used the Pytorch Metric Learning (PML) (Musgrave et al. (2020b)) library.

For all experiments we use the supervised contrastive loss (Khosla et al. (2020)) with a pair margin miner, where the default values are used, *i.e.*, *pos_margin*=0.2 and *neg_margin*=0.8 (Musgrave et al. (2020b)). Default data augmentations in the TIMM library are used along with RandAugment (Cubuk et al. (2020)) profile *rand-m9-mstd0.5*, *reprob* is set to 0.2 and mixed precision training. The embeddings are L2-normalized and cosine similarity is used as the distance metric for training and evaluation. We used AdamW optimizer (Loshchilov & Hutter (2017)) with initial learning rate 0.0001 and the learning rate is multiplied by 0.33 at every 15 epochs. We train for a total of 50 epochs and the best model with respect to Recall@1 on the validation set is selected. We used a custom sampler to ensure that there are four images per class in each minibatch.

To ensure that the feature embeddings are stable, for all methods, we employ a pre-warmup stage to finetune the randomly initialized last layer[4] which projects the backbone features to a 512 dimensional embedding. Specifically, we finetune this layer for 2 epochs with the standard supervised constrastive loss. For this stage, SGD with the learning rate of 0.001 is used.

For our AXBN method, the noise hyperparameters are set as follows: $q = 1$, $p_0 = 1$, and $r = 0.01$. We did a small grid search on the SOP dataset for batch size 64 to obtain $r$ and other parameters are not tuned. The same value of $r$ is used for all datasets and all batch sizes. In addition to this, we also tune the interval at which to update the Kalman gain (Eq. (20)). We found the value 100 to work well for SOP and IN-SHOP, and the value 10 is used for DEEPFASHION2. While it may possible to tune these hyperparameters for each setting individually to squeeze out a bit more performance, we did not do that in our experiments. All our experiments are performed on a V100 GPU in the AWS cloud.

As noted earlier, our technique is a simple modification to the XBM approach and we implemented it as a custom loss in the PML library. Our code will be released upon publication.

### 5.2 Datasets

We evaluated on three large-scale datasets for few-shot image retrieval following the standard protocol.

**Stanford Online Products.** Stanford Online Products (SOP) (Oh Song et al. (2016)) contains 120,053 online product images in 22,634 categories. There are only 2 to 10 images for each category. Following (Oh Song

---

[4]Note that the rest of the network is pretrained on ImageNet.

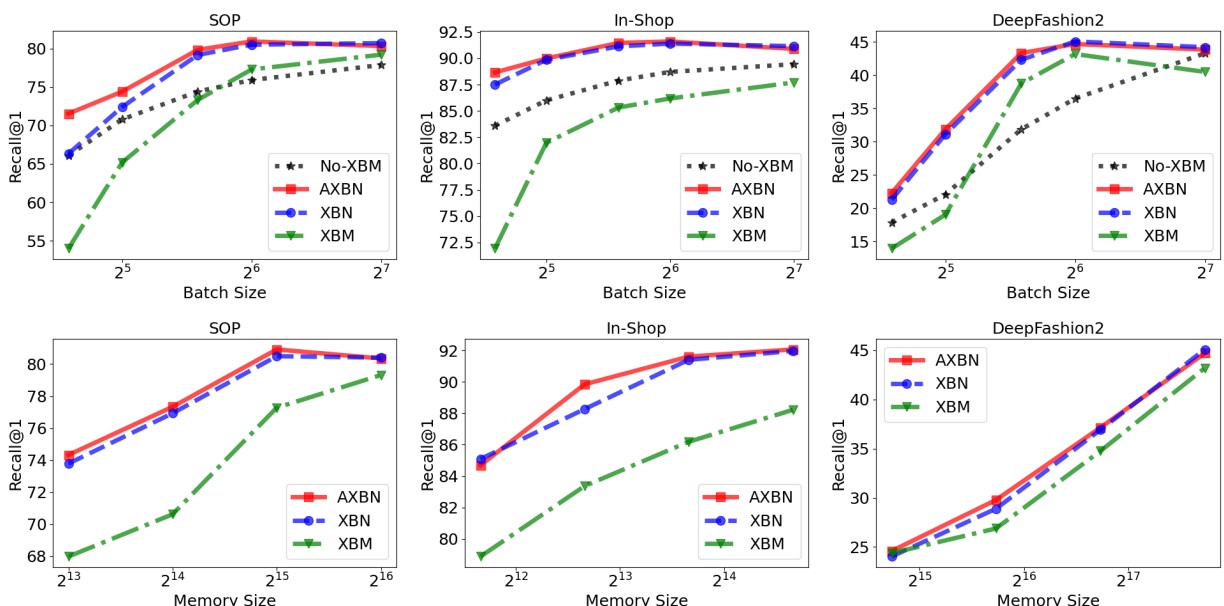

Figure 1: **Top:** *Recall@1* vs. *batch size where cross batch memory size is fixed to* 50% *(SOP and IN-SHOP) or* 100% *(DEEPFASHION2) of the training set.* **Bottom:** *Recall@1* vs. *cross batch memory size with batch size is set to* 64*. In all cases, our algorithms significantly outperform XBM and the adaptive version is better than the simpler XBN method especially for smaller batch sizes.*

| Algorithm | SOP | | IN-SHOP | | DEEPFASHION2 | |
|---|---|---|---|---|---|---|
| | Recall@1 | Recall@10 | Recall@1 | Recall@10 | Recall@1 | Recall@10 |
| No-XBM | $75.94 \pm 0.03$ | $89.74 \pm 0.05$ | $88.76 \pm 0.15$ | $97.76 \pm 0.13$ | $36.45 \pm 0.06$ | $61.44 \pm 0.31$ |
| XBM | $76.80 \pm 2.65$ | $89.34 \pm 1.90$ | $86.17 \pm 0.18$ | $96.58 \pm 0.28$ | $41.22 \pm 2.96$ | $62.24 \pm 3.54$ |
| Ours XBN | $\underline{80.62} \pm 0.17$ | $\underline{91.85} \pm 0.11$ | $\underline{91.49} \pm 0.07$ | $\underline{98.31} \pm 0.10$ | $\underline{45.12} \pm 0.08$ | $\mathbf{66.32} \pm 0.17$ |
| Ours AXBN | $\mathbf{80.73} \pm 0.30$ | $\mathbf{91.98} \pm 0.15$ | $\mathbf{91.51} \pm 0.20$ | $\mathbf{98.35} \pm 0.09$ | $\mathbf{45.33} \pm 0.28$ | $\underline{66.26} \pm 0.63$ |

Table 1: *Summary of results for a particular setting where batch size is* 64 *and the cross batch memory size is* 50% *(SOP and IN-SHOP) or* 100% *(DEEPFASHION2) of the training set. The experiments are repeated three times and the mean and standard deviation are reported. Best numbers are in bold and the second best numbers are underlined. In all cases, our methods significantly outperform XBM. The larger standard deviations for XBM indicate its training instability.*

et al. (2016)), we use 59,551 images (11,318 classes) for training, and 60,502 images (11,316 classes) for testing.

**In-shop Clothes Retrieval.** In-shop Clothes Retrieval (IN-SHOP) (Liu et al. (2016)) contains 72,712 clothing images of 7,986 classes. Following (Liu et al. (2016)), we use 3,997 classes with 25,882 images as the training set. The test set is partitioned to a query set with 14,218 images of 3,985 classes, and a gallery set having 3,985 classes with 12,612 images.

**Deep Fashion 2.** Deep Fashion 2 (DEEPFASHION2) (Ge et al. (2019)) consumer to shop retrieval dataset contains 217,778 cloth bounding boxes that have valid consumer to shop pairs. We use ground truth bounding boxes in training and testing. We follow the evaluation protocol described in (Ge et al. (2019)) and use the validation set for evaluation. It has 36,961 items in the gallery set and 12,377 items in the query set. We consider a pair is unique if the *pair_id*, *category_id*, and *style* match, following the original paper. To this end, there are about 31,517 unique labels in the training set, 7,059 labels in the gallery set, and 3,128 labels in the query set. Note that, in contrast to the other two datasets, DEEPFASHION2 is much larger and there is a domain gap between the images in query and gallery sets.

| | Algorithm | SOP | | IN-SHOP | |
|---|---|---|---|---|---|
| | | Recall@1 | Recall@10 | Recall@1 | Recall@10 |
| | No-XBM | $75.94 \pm 0.03$ | $89.74 \pm 0.05$ | $88.76 \pm 0.15$ | $97.76 \pm 0.13$ |
| | No-XBM$^{512}$ | $79.63 \pm 0.13$ | $91.76 \pm 0.02$ | $90.39 \pm 0.01$ | $98.18 \pm 0.03$ |
| | XBM | $76.80 \pm 2.65$ | $89.34 \pm 1.90$ | $86.17 \pm 0.18$ | $96.58 \pm 0.28$ |
| | XBM* | $79.53 \pm 0.09$ | $91.64 \pm 0.01$ | $90.91 \pm 0.10$ | $98.20 \pm 0.03$ |
| Ours | XBN | $\underline{80.62} \pm 0.17$ | $\underline{91.85} \pm 0.11$ | $\underline{91.49} \pm 0.07$ | $\underline{98.31} \pm 0.10$ |
| | AXBN | $\mathbf{80.73} \pm 0.30$ | $\mathbf{91.98} \pm 0.15$ | $\mathbf{91.51} \pm 0.20$ | $\mathbf{98.35} \pm 0.09$ |

Table 2: *Additional comparisons on smaller datasets where batch size is* 64 *and memory size is* 50% *of the training set. Here, No-XBM$^{512}$ denotes No-XBM with batch size* 512 *and XBM\* denotes adding the loss on the minibatch to the XBM loss to stabilize training. The experiments are repeated three times and the mean and standard deviation are reported. Best numbers are in bold and the second best numbers are underlined. In all cases, our methods clearly outperform both versions of XBM.*

### 5.3 Results

Following the paradigm of Musgrave et al. (2020a), we compare our approach against the original XBM method and the one without any cross batch memory while keeping everything else the same for fair comparison. In this way, we can clearly demonstrate the benefits of expanding the reference set (*i.e.*, cross batch memory) and adapting it to ensure that it is up to date.

In Fig. 1, we plot the performance of different methods by varying the batch size and varying the cross batch memory size. We can clearly see that our methods significantly outperform XBM in all cases, validating our hypothesis that it is important to ensure that the reference set embeddings are up to date. Furthermore, while our simpler XBN method is powerful, our adaptive method yields slightly better performance for the smaller minibatches where the sampling noise is higher.

In Table 1, we summarise results for a particular setting with batch size 64 and reference set size is 50% of the training set for SOP and IN-SHOP and 100% for DEEPFASHION2. We repeat the experiments three times and report the mean and standard deviation of the results. In all cases, our methods significantly outperform XBM, confirming the merits of tackling representational drift. Furthermore, the XBM results have large standard deviations for repeated experiments, which we believe indicates its training instability due to outdated embeddings.

#### 5.3.1 Additional Comparisons

On the smaller SOP and IN-SHOP datasets we perform further experiments to understand the benefits of a larger and up to date reference set. To this end, in Table 2 we provide the results of No-XBM version with a larger minibatch size. Even with the 8× larger minibatch, the performance is worse than our approach. Note that, larger minibatch size significantly increases the required GPU memory, whereas storing the embeddings adds little overhead (Wang et al. (2020)).

Furthermore, we include the performance of a modified XBM method where the XBM loss is summed with the loss on the minibatch to stabilize training. This is a trick used in the original XBM code[5] but was not mentioned in the paper. While this helps the XBM performance, it is still worse than both of our methods. Clearly this trick diminishes the value of using a larger reference set and confirms our hypothesis that it is necessary to ensure that the reference set embeddings are up to date.

In the plots and table above, XBM is sometimes worse than not using the cross batch memory. We believe this is due to outdated embeddings providing inaccurate supervision signal. Note that, when the loss on minibatch is added to the XBM loss, performance improves and the standard deviation of repeated experiments decreases. For our methods, we did not observe any improvements when adding the minibatch loss.

More experiments comparing other normalization techniques and feature drift diagnostics are provided in the appendix.

---

[5]https://github.com/msight-tech/research-xbm

## 6 Discussion

In this paper, we have introduced an efficient approach that adapts the embeddings in the cross batch memory to tackle representational drift. This enables us to effectively expand the reference set to be as large as the full training set without significant memory overhead. Our simple XBN approach significantly outperforms the standard XBM method, demonstrating the significance of ensuring that the reference set embeddings are up to date. Furthermore, we introduced an adaptive version (named AXBN) which uses Kalman filter to model the noisy estimation process, performs slightly better than the simpler XBN method for smaller batch sizes. We believe our technique will become part of the toolset for metric learning due to its simplicity and effectiveness. Indeed, other applications where accumulation of examples of many minibatches is needed may also benefit from our approach.

### 6.1 Limitations and Societal Impact

In our XBN method, we only consider a linear transformation function to tackle representational drift. Nevertheless, as noted earlier, it is possible to incorporate higher-order terms and/or cross-correlation among the elements of the embeddings. Additionally, in our AXBN method, it is not clear how to choose the Kalman filter noise variances as they are problem dependent. Currently we treat them as hyperparameters. However, it would be interesting to come up with an automatic mechanism to obtain these variances depending on the dataset and architecture. We also made some simplifying assumptions for this method such as independence among the dimensions for computational benefits. While our resulting method works well in our experiments, the full potential of our approach may be limited due to these assumptions.

Our method focuses on improving the learning effectiveness in metric learning by ensuring the cross batch memory embeddings are up to date. As with any method that optimises for performance on the training data, our method may amplify dataset bias, however, methods that combat dataset bias may also benefit from the better training signal. Overall, our method may contribute to the societal impact of deep metric learning, both positive and negative.

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

| Algorithm | L2 + BN | | L2 + MBN | | L2 + LN | | L2 | |
|---|---|---|---|---|---|---|---|---|
| | Recall@1 | Recall@10 | Recall@1 | Recall@10 | Recall@1 | Recall@10 | Recall@1 | Recall@10 |
| No-XBM | 48.96 | 65.06 | 48.96 | 65.06 | 75.77 | 89.54 | 75.94 | 89.74 |
| XBM | 48.96 | 65.06 | 48.96 | 65.06 | 79.51 | 91.17 | 76.80 | 89.34 |
| XBN | 48.96 | 65.06 | 48.96 | 65.06 | **80.58** | **91.77** | **80.62** | **91.85** |

Table 3: *Experiments with BN, MBN, and LN for emebedding normalization and compared against our method on the SOP dataset. While networks with BN and MBN yield poor results, LN improves over the standard XBM method. Nevertheless, our approach (XBN) improves further and the best performance of our method is attained when only L2 is used for embedding normalization. Note, the best results for BN and MBN are achieved during the warmup stage and hence identical.*

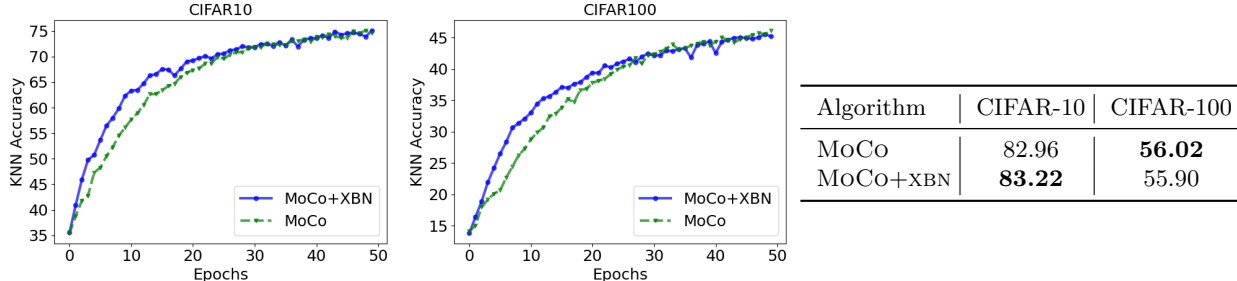

Figure 2: *MOCO with and without our XBN approach. The left two figures show the the training curves for the first 50 epochs and the right table shows the final k-NN accuracy after 200 epochs. XBN clearly improves the training of MOCO early in the training demonstrating the benefit of adapting stored embeddings even when a momentum encoder is used.*

Hongwei Yong, Jianqiang Huang, Deyu Meng, Xiansheng Hua, and Lei Zhang. Momentum batch normalization for deep learning with small batch size. In *European Conference on Computer Vision*, pp. 224–240. Springer, 2020.

Zhun Zhong, Liang Zheng, Zhiming Luo, Shaozi Li, and Yi Yang. Invariance matters: Exemplar memory for domain adaptive person re-identification. In *Proceedings of the IEEE/CVF Conference on Computer Vision and Pattern Recognition*, pp. 598–607, 2019.

## A  Comparison to other Normalizations

We experiment with Batch Normalization (BN) (Ioffe & Szegedy (2015)), Momentum Batch Normalization (MBN) (Yong et al. (2020)), and Layer Normalization (LN) (Ba et al. (2016)) to normalize the feature embeddings and the results on the SOP dataset with batch size 64 are reported in Table 3. Here, the nomalization approach is preceded by L2 normalization to mimic the protocol of our approach. For BN and MBN the best validation performance is obtained during the warmup stage and we found the networks with BN/MBN for embedding normalization untrainable. While networks with BN and MBN yield poor results, LN improves over the standard XBM method. Nevertheless, our approach (XBN) improves further and the best performance of our method is attained when neither BN nor LN is used for embedding normalization.

Even though we are unaware of any work that used BN to normalize feature embeddings, we used BN for embedding normalization in various settings (with/without L2 normalization, freezing/not freezing BN parameters, and with/without tracking running statistics) and the resulting networks were untrainable for a range of learning rates and attained poor results. We hypothesize that BN is not designed to be used for embedding normalization and it would require deeper analysis and modifications to it if one wants to use it. Similar behaviour is observed even for MBN where we used the recommended hyperparameters and tested with two batch sizes: 64 (with $m_0 = 128$) and 16 (with $m_0 = 32$). In both the settings the behaviour is similar to BN.

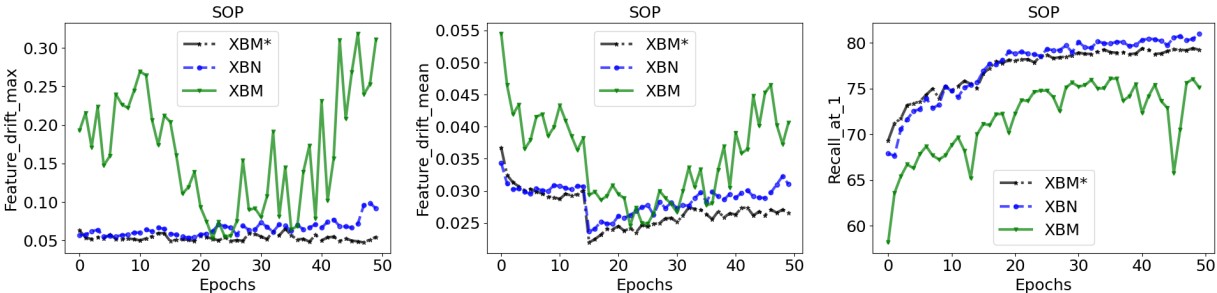

Figure 3: *Maximum and mean feature drift on a minibatch of samples on the SOP dataset for XBM and our XBN method, and the corresponding performance on the validation set. Our method has much smaller feature drift for the most part of the training compared to XBM and yields improved performance on the validation set.*

## B  MoCo with XBN

MoCo He et al. (2020) is a self-supervised learning approach that uses an external memory similar to XBM and uses a momentum based encoder to handle representational drift. To understand if our approach is beneficial despite the presence of the momentum encoder, we experimented MoCo on CIFAR-10/100 datasets with our XBN approach to adapt the stored embeddings. Specifically, we followed the MoCo example provided in PML[6] including all hyperparameter values except the batch size, which is set 128.

As shown in Fig. 2, our approach indeed improves the training of MoCo especially early in the training (*e.g.*, $\sim 4\%$ improvement at epoch 15 for both datasets) where features are adapted quickly. Note this improvement is on top of the momentum encoder where the momentum parameter is set to 0.99. Even though the gap is reduced towards the end of training, this clearly demonstrates the benefits of our approach outside of metric learning.

## C  Feature Drift Diagnostics

We measure feature drift (refer to Eq. (5)) for XBM and our XBN method for the SOP dataset in Fig. 3. We fix $\Delta k = 1$ and measure the drift on a minibatch of training samples with batch size 64. The maximum or mean drift is computed on the minibatch and the respective quantities are averaged over the epoch. Our method has much smaller feature drift for the most part of the training compared to XBM and yields improved performance on the validation set.

Note that, XBM* also reduces feature drift similar to our method, however, its performance is inferior to our approach as reported in Table 2 and Fig. 3. We hypothesize that, even though XBM* shows slow-drift, adding minibatch loss diminishes the value of using a larger reference set, leading to inferior performance.

As opposed to our approach which directly tackles feature drift, the slow-drift phenomenon is emergent for XBM* (also for No-XBM as shown in Wang et al. (2020)), and the reason is not well understood. We believe relying on a principled approach to handle feature drift is valuable and as shown in the experiments our approach yields superior performance.

## D  Large Model Experiments

All the experiments in the main paper are provided with ResNet-50 architecture. Here, we experiment with much larger SwinTransformer (Liu et al. (2021)). Similar to other experiments in the paper ImageNet pretrained weights are used to initialize the network. The results are provided in Table 4. The behaviour is similar to ResNet-50 and our approaches clearly outperform the original XBM. However, even with small batch sizes the improvement due to the adaptive version is marginal. We hypothesize that these large models are

---

[6]https://github.com/KevinMusgrave/pytorch-metric-learning/blob/master/examples/notebooks/MoCoCIFAR10.ipynb

---

**Algorithm 1** AXBN update at iteration $k$

---

**Require:** $\bar{\mathcal{R}}^{k-1}, \bar{\mathcal{B}}^k, \hat{\boldsymbol{\mu}}_{k-1}, \hat{\boldsymbol{\sigma}}_{k-1}, K_{k-1}, p_{k-1}, q, r$      ▷ Reference set, current batch, and state variables
**Ensure:** $\bar{\mathcal{R}}^k, \hat{\boldsymbol{\mu}}_k, \hat{\boldsymbol{\sigma}}_k, K_k, p_k$      ▷ Updated reference set and state variables
 1: $p_{k,k-1} \leftarrow p_{k-1} + q$      ▷ Predicted noise estimate
 2: $K_k \leftarrow p_{k,k-1} / (p_{k,k-1} + r)$      ▷ Kalman gain
 3: $p_k \leftarrow (1 - K_k) \, p_{k,k-1}$      ▷ Updated noise estimate
 4: $\hat{\boldsymbol{\mu}}_k \leftarrow \hat{\boldsymbol{\mu}}_{k-1} + K_k \left( \mathbb{E}\left[\bar{\mathcal{B}}^k\right] - \hat{\boldsymbol{\mu}}_{k-1} \right)$      ▷ Mean estimate
 5: $\hat{\boldsymbol{\sigma}}_k \leftarrow \hat{\boldsymbol{\sigma}}_{k-1} + K_k \left( \sigma\left[\bar{\mathcal{B}}^k\right] - \hat{\boldsymbol{\sigma}}_{k-1} \right)$      ▷ Variance estimate
 6: $\mathbf{z}^k \leftarrow \left( \mathbf{z}^{k-1} - \mathbb{E}\left[\bar{\mathcal{R}}^{k-1}\right] \right) \hat{\boldsymbol{\sigma}}_k / \sigma\left[\bar{\mathcal{R}}^{k-1}\right] + \hat{\boldsymbol{\mu}}_k \,,$     for each $(\mathbf{z}^{k-1}, y) \in \bar{\mathcal{R}}^{k-1}$      ▷ Normalization
 7: $\bar{\mathcal{R}}^k \leftarrow \{(\mathbf{z}^k, y)\}$      ▷ Store the updated reference set

---

| | Algorithm | Batch size = 16 | | Batch size = 32 | |
|---|---|---|---|---|---|
| | | Recall@1 | Recall@10 | Recall@1 | Recall@10 |
| | No-XBM | 77.80 | 90.70 | 81.81 | 92.91 |
| | XBM | 61.18 | 76.95 | 86.99 | 95.39 |
| Ours | XBN | 86.25 | 94.89 | 87.59 | 95.57 |
| | AXBN | **86.40** | **94.96** | **87.74** | **95.66** |

Table 4: *SwinTransformer results on the SOP dataset with batch size 16 and 32. As expected, in both cases our approaches clearly outperform XBM and our adaptive version AXBN is slightly better than the simple XBN method.*

stable pretrained models and therefore the noise in minibatch based feature statistics is small, and Kalman filter based noise estimation does not improve significantly.

## E   More on the AXBN Method

Since we have made several simplifying assumptions to the Kalman filter based noise estimation, we provide the update in its simplest form in Algorithm 1.

Note that, Exponential Moving Average (EMA) can be thought of as a special case of our final AXBN approach where the Kalman gain is replaced with a constant throughout training. To this end, we perform an experiment with EMA on the SOP dataset with batch size 64 using ResNet-50 architecture. We varied the momentum parameter within the range $\{0.1, 0.2, ..., 0.9\}$ (where 0 corresponds to XBN) and the best performance is obtained when the momentum parameter is 0.1. The results are reported in Table 5.

| Algorithm | SOP | |
|---|---|---|
| | Recall@1 | Recall@10 |
| XBN | 80.62 | 91.85 |
| XBN+EMA | 80.48 | 91.95 |
| AXBN | **80.73** | **91.98** |

Table 5: *Results with EMA version of our approach. The results are competitive to both of our methods but estimating the Kalman gain performs slightly better.*

As discussed in Section 5.3, the AXBN approach is useful when the sampling noise due to minibatches is high. There are two factors that affect the sampling noise: 1) minibatch size, and 2) the stability of the embedding network. The adaptive method may not be necessary when one has a stable embedding network (a network that shows slow feature-drift) and/or the minibatch size is large.

| | Algorithm | SOP | |
|---|---|---|---|
| | | Recall@1 | Recall@10 |
| | No-XBM | 78.69 | 91.21 |
| | XBM | 78.38 | 90.39 |
| Ours | XBN | 79.96 | 91.02 |
| | AXBN | **80.31** | **91.31** |

Table 6: *Results with batch size* 256. *The gap between different methods decreased compared to smaller batch sizes.*

## F   Larger Batch Sizes

We performed an experiment with batch size 256 on the SOP dataset (maximum batch size allowed by our A10G GPU with 25GB memory) and the results are reported in Table 6. From these results and Fig. 1 top-row in the main paper, one may extrapolate that larger batch sizes tend to lead to smaller gap between the methods. This is expected as when the batch size is large enough, cross batch memory may not be required.

Nevertheless, the point of our paper is not about when to use cross batch memory (which has already been established in (Wang et al. (2020))), rather in cases where cross batch memory is relevant, our approach is the most effective way to use it.

