# OpenReview forum: "Adaptive Cross Batch Normalization for Metric Learning"
_TMLR — Rejected by TMLR_

### Review · Reviewer_o42X · 2023-01-08

**Summary Of Contributions:**

This paper aims to study the batch normalization technique for metric learning with ranking loss. This paper pointed out that the method with the ranking loss highly depends on the minibatch size. The proposed approach is based on the scheme of XBN, accumulating the embeddings across minibatche (Wang et al. [2020]), and aims to solve the representational drift issue between the accumulated embeddings and the feature embeddings at the current training iteration as the learnable parameters are being updated. The proposed approach, called adaptive cross batch normalization, adapts the stored embeddings to match the first and second moments of the current embeddings at each training iteration. Experiments on image retrieval datasets justified the effectiveness of the proposed approach.

**Audience:**

No

**Claims And Evidence:**

Yes

**Requested Changes:**

Please address the weaknesses.

(1) Compare to momentum batch normalization.

(2) Explain the results in Table 1 and Table 2.

(3) Describe the implementation of XBM and how the results are obtained.

(4) Report the results of large batch size.


**Strengths And Weaknesses:**

Strengths:
(1) This writing is very good. The concepts, including triplet loss, cross batch normalization and its justification, and Kalman filter for statistics estimation, are formally described.

(2) The proposed approach is mathematically formulated clearly.

(3) The experimental results justified the proposed approach is helpful.


Weaknesses
(1) There are some other batch normalization schemes, for example momentum batch normalization [1]. It would be better to discuss and compare this scheme.


(2) From the results in Table 1 and Table 2, the improvement for smaller datasets (Table 2) over XBM is not as significant as Table 1.

(3) It is unclear how the implementation of XBM and the result is obtained.

(4)  Does a larger batch size lead to good performance?

[1] Hongwei Yong, Jianqiang Huang, Deyu Meng, Xian-Sheng Hua, Lei Zhang: Momentum Batch Normalization for Deep Learning with Small Batch Size. ECCV (12) 2020: 224-240)

---

> ### Author Response · Authors · 2023-01-25
> **Response to o42X**
>
> We thank the reviewer for appreciating that the paper is clearly written and the experiments justify the merits of our approach. We address the specific concerns below and we have updated the manuscript accordingly.
>
> ### Momentum Batch Normalization (MBN)
>
> * Thanks for suggesting the MBN paper (Yong et al., 2020), we will cite it in the revised manuscript. Even though the momentum approach is relevant, it is shown by the authors to be effective in the small batch regime (<16 batch sizes) and our experiments are conducted with batch size 64. Nevertheless, we performed an experiment with MBN as a feature normalization approach in Appendix A with batch size 64 (with $m_0=128$) and 16 (with $m_0=32$) while the other parameters set according to the MBN paper. The results are similar to batch normalization (BN) in the sense that, the resulting networks are untrainable and attained poor results. We hypothesize that BN/MBN is not designed to be used for embedding normalization and it would require deeper analysis and modifications to it if one wants to use it.
> * Additionally, we would like to clarify that one can use momentum based feature statistics to normalize the cross batch embeddings. This approach is a special case of our Kalman filter based method (AXBN, also noted by reviewer jxoW) where rather than estimating the Kalman gain one can set it to a fixed value or a predefined schedule. Please see the response to reviewer jxoW for discussion on experiments that we conducted with the momentum based approach.
>
> ### Table 1 vs Table 2
>
> * The results compared to XBM is significant in both the tables. In Table 2 we additionally show XBM* which improves XBM but is still worse than our proposed approach. As discussed in Sec 5.3.1, XBM* is a trick used in the original code where the minibatch loss is added to the XBM loss to stabilize training. This trick was not mentioned in the XBM paper but is nevertheless important for obtaining their reported results.
>
> ### XBM implementation
>
> * We used the XBM implementation from the PyTorch Metric Learning library (Musgrave et al. (2020b)), which we now mention in our revised paper. As noted in Sec. 5.1, for all methods, the model with the best Recall@1 on the validation set is chosen.
>
> ### Larger batch size
>
> * We performed an experiment on the SOP dataset with batch size 256 (maximum batch size allowed by our A10G GPU with 24GB memory) and the results are reported below. These results and plots in Fig. 1 top-row show that the performance of all the methods improve (or saturate) with larger batch sizes and the gap between the methods tends to decrease. It is intuitive that larger batch size may reduce the significance of using cross batch memory and this may be extrapolated from these results. Nevertheless, No-XBM with 512 batch size is still worse than our method with batch size 64 (Table 2) showing the usefulness of cross batch memory.
>
> * We would like to point out that our paper is not about when to use cross batch memory (which has already been established in (Wang etal. 2020)), rather in cases where cross batch memory is relevant, our approach is the most effective way to use it. We have added this discussion in Appendix F.
>
> | Method	    | Recall@1	  | Recall@10 |
> |--------------|-------------|-----------|
> | No-XBM	    | 78.69	      | 91.21	  |
> | XBM	        | 78.38	      | 90.39	  |
> | XBN	        | 79.96	      | 91.2	  |
> | AXBN	        | **80.31**	  | **91.31** |

---

> > ### Comment · Reviewer_o42X · 2023-02-18
> > **My comments were well handled**
> >
> > Thanks the authors for the feedback! My comments were well addressed.
> >
> > It is fine to accept this paper though I think that the novelty is not very significant.

---

### Review · Reviewer_jxoW · 2023-01-15

**Summary Of Contributions:**

Metric learning often relies on the comparison of embeddings of each sample to a reference set of samples, and prior work has shown that extending the reference set beyond the current batch (by keeping the embeddings of prior batches in memory) can greatly enhance metric learning performance. This paper introduces a Normalization procedure to match the statistics of the reference set (obtained in prior iterations) to the statistics of the current model’s output embeddings. This idea has a very simple and intuitive implementation (resembling that of other normalization procedures like BN, LN, etc), but yields significant performance gains in several image retrieval tasks. Beyond normalization of the reference set, the paper also introduces Kalman filter procedure to estimate the statistics of the current model embeddings from noisy estimates. The adaptive normalization procedure was shown useful when batch size is small, since, in this case, the sample mean and std within the batch are poor estimators of the true mean and std. Although the adaptive normalization shows no improvement when batch size is large, it could be potentially useful to train large models.

**Audience:**

Yes

**Broader Impact Concerns:**

Broader impacts have been well address by the authors. I have no other concerns.

**Claims And Evidence:**

Yes

**Requested Changes:**

I would like to see all points raised in the weaknesses section addressed (perhaps except for the 4th point).

**Strengths And Weaknesses:**

**Strengths:**

- The main contributions/claims of the paper are two-fold. 1) Reference set normalization is required when training metric learning systems that require a memory-based reference set. 2) Adaptive normalization is helpful when available estimates of embeddings statistics are (unbiased but) noisy. Both of these contributions have been well-validated experimentally. Performance gains are significant and consistent across all datasets.
- The proposed normalization procedure is also quite simple and effective, which, in my view, is a major strength, and it would likely lead to high adoption.
- Reliable experimental settings. Experiments are conducted in relatively large datasets, averaged over three random seeds. The consistent improvements provide strong evidence for the method’s advantages.
- The paper is overall well-written and easy to follow. See weakness (Kalman filter comment) for something that could be improved in this regard.

**Weaknesses:**

- Although effective, the normalization procedure is quite simple. If indeed novel, then I would have no reservations recommending acceptance. However, since I’m not an expert in metric learning, I do not know whether feature normalization is commonplace in other metric learning procedures. The authors did not address this question in related work, as well. Hence, I will leave the assessment of novelty to other reviewers with deeper expertise in this specific topic.
- Missing realistic experiments where low-batch sizes are required, in order to highlight the benefit of the adaptive procedure. One potential scenario where this would happen is when tuning large models, eg with large ViT models, or large convnets like regnet.
- The mathematical treatment of Kalman filter is not satisfactory. Eq 18 and 21 contradict each other (same with 19 and 22). For example, u_k in eq 18 should be the state estimate given all observations up to time t-1 (since there are no additional inputs), which is not clear. Also, since there are no controlled inputs (ie state transition model is the identity), the Kalman filter simplifies greatly into u_k =  (I-K_k) u_{k-1} + K_k v_k. This is just an exponential moving average with factor K_k that approaches 1 over time. This is not clear in the paper, making the method appear more complex than what it actually is. In my opinion, the simplification (including feature independence) should be carried out to its simpler form, and an algorithm should be provided (in its simplest form).
- Although not critical for acceptance, I would like to see a comparison to a simple EMA (exponential moving average) update of the embedding statistics, as a simpler alternative to the Kalman filter. From my understanding, this would be similar, except for the Kalman gain K_k which would remain constant throughout training.
- References: Wikipedia is cited twice. Although a great resource, Wikipedia should not be used as a reference, as it can be continuously updated. Please use better references, e.g. a reference book. Also, add a reference to Kalman filter.

---

> ### Author Response · Authors · 2023-01-25
> **Response to jxoW**
>
> We thank the reviewer for recognizing the simplicity of our method as a major strength and confirming that the experimental results validate the effectiveness of our approach. We address specific comments below and note that our manuscript has been updated to incorporate the reviewer’s suggestions where applicable.
>
> ### Novelty of our normalization
>
> * Normalizing each feature instance-wise (e.g., L2 normalization) is common in metric learning. However, using minibatch (or cross batch memory) statistics to normalize features is novel. We introduce it for the specific purpose of addressing feature drift in the XBM approach. As shown in Appendix A, directly using other normalizations (e.g., batch normalization or layer normalization) for this purpose is not as effective as our approach.
>
> ### Large model experiments
>
> * We thank the reviewer for mentioning large-scale experiments where smaller batch sizes are required. We performed an experiment using the SwinTransformer (Liu et al. 2021) on the SOP dataset with batch size 16 and 32 and the results are reported below. Similar to other experiments in the paper, we initialize the network using ImageNet pretrained weights. The behaviour is similar to Resnet-50 in that, the performance gap between AXBN and XBN is marginal. We hypothesize that these large models are stable pretrained models and therefore the noise in minibatch based feature statistics is small, and Kalman filter based noise estimation does not improve significantly. Nevertheless, a noticeable gain between the original XBM and our XBN approach is observed. We added these results to Appendix D.
> * Due to the large-size of SwinTransformer and comparatively smaller SOP dataset, we were unable to train the model from scratch.
>
> | Batch size	 | 16	           | 16	         | 32          | 32          |
> |-------------|---------------|-------------|-------------|-------------|
> | __Method__	 | __Recall@1__	 | __Recall@10__	  | __Recall@1__	   | __Recall@10__	  |
> | No-XBM	     | 77.8 	        | 90.7	       | 81.81       | 92.91	      |
> | XBM	        | 61.18	        | 76.95	      | 86.99	      | 95.39	      |
> | XBN	        | 86.25	        | 94.89	      | 87.59 	     | 95.57	      |
> | AXBN	       | __86.4__	     | __94.96__	  | __87.74__	  | __95.66__	  |
>
> ### Kalman filter description
>
> * We thank the reviewer for pointing out the issue with the Kalman filter equations. To improve clarity, we now introduce notation to denote the intermediate state and variance estimates. The Eq. (18 - 22) have been updated to reflect this.
> * We also provide the AXBN update in its simplest form in Appendix E.
> * We have updated the references.
>
> ### EMA based normalization
>
> * We thank the reviewer for identifying EMA as a special case of our AXBN approach. We performed an experiment on the SOP dataset with Resnet-50 by varying the momentum parameter within the range {0.1, 0.2, ..., 0.9} (0 corresponds to XBN). The best performance is obtained when the momentum parameter is 0.1. We now include these results in Appendix E.
>
> | Method	  | Recall@1	 | Recall@10	 |
> |----------|-----------|------------|
> | XBN	     | 80.62	    | 91.85	     |
> | XBN+EMA	 | 80.48	    | 91.95	     |
> | AXBN	    | __80.73__	 | __91.98__	 |

---

### Review · Reviewer_SGzu · 2023-01-21

**Summary Of Contributions:**

In this paper, the authors proposed to leverage Kalman Filter to estimate the statistics used in cross-batch normalization for enlarging the sample pool when calculating ranking loss, in the training process of a metric space. The effectiveness of the model is empirically shown by comparing the proposed method and existing way of directly estimating the statistics without considering the noise issue in the mini-batch training setting.

**Audience:**

Yes

**Claims And Evidence:**

No

**Requested Changes:**

Please check the cons for details

**Strengths And Weaknesses:**

Pros:
1. The reviewer really enjoys the idea of the paper.
2. The provided empirical results are strong.

Cons:
1. There are two vital gaps that are not justified sufficiently in the current manuscript.
- a. The first gap is that the authors accused the existing method of possibly changing the rank dramatically. But the authors didn't provide any justification on whether the proposed method has less rank-changing issues either empirically or theoretically.
- b. It is not clear whether the performance gain comes from better estimation of the statistics or not. More in-depth analysis should have been provided, e.g., 1) controlling the noise level of the estimation through combing the noisy plain estimation and the proposed estimation and see whether there is certain negative correlation between the noisy level and the final performance; 2) It is also possible to control the number of samples that have a wrong rank because of the estimation of the feature vectors and check the relationship between this and the final performance.

2. The current evaluation setting does not consider a true validation set. Therefore it is not clear whether the performance gain comes from fitting the test set through hyper-parameter tuning or not. More scientific dataset split method is necessary.

---

> ### Author Response · Authors · 2023-01-25
> **Response to SGzu**
>
> We thank the reviewer for expressing the enjoyment while reading the paper and confirming that the empirical results are strong. We address specific comments below and have incorporated suggestions into the revised manuscript.
>
> ### Feature drift diagnostics
>
> * We understand that the reviewer’s major concern is whether the improvement of our method is due to solving the feature-drift issue. We answer this below.
> * The main argument we put against the original XBM paper is that the feature-drift (Eq. 5) is not handled during training. We tackle this via our normalization scheme (Eq. 6). We compare the feature-drift diagnostics during training of original XBM and our XBN in Fig. 3 in the appendix. As demonstrated in the figure, our approach has very small feature-drfit throughout training and yields significantly better performance.
> * We would like to clarify that, the difference between XBM and XBN in our experiments is **only** the normalization scheme (Eq. 6) and we keep everything else fixed for fair comparison (see Sec. 5.3). Therefore, the improvements of our method can only be attributed to the normalization scheme which is designed to handle feature-drift.
>
> ### Dataset splits
>
> * We recognize the reviewer’s concern that the validation and test splits are the same. This split is not designed by us and we simply use the data-splits recommended by the source dataset papers and the papers that used these datasets including the original XBM paper. Unfortunately, it is impractical to repeat all the experiments with different splits during the rebuttal period.
> * Nevertheless, we would like to clarify that, the gains of our methods are **not** the result of fitting to the test set. Because, for our XBN approach, there are **no additional hyperparameters** and all the hyperparameters are the same between XBN and XBM as mentioned in Sec. 5.3. Furthermore, the learning rate and other standard hyperparameters were tuned with a fixed batch size and the same setting was used across different batch sizes and cross batch memory sizes in Fig. 1.
> * We believe, this (and the feature-drift diagnostics above) clearly demonstrate that the gains of our approach are not merely due to hyperparameter tuning and our method indeed improve feature-drift and consequently yields better performance.

---

### Review · Reviewer_e7Xy · 2023-01-21

**Summary Of Contributions:**

The paper discussed the metric learning in the context of image retrieval, where the central task is to learn an embedding space that reflects the clustering structure represented by the labels. Thus, the fact that the embedding space need to be updated along the training process becomes essential in comparison to other tasks. The authors introduce a conceptually very intuitive idea to address this issue and achieves reasonably good empirical performances.

**Audience:**

Yes

**Broader Impact Concerns:**

none noted.

**Claims And Evidence:**

Yes

**Requested Changes:**

While the paper is nicely written with an interesting idea, there are many more questions needed to be answered, for examples

   - The method introduces several assumption and component, and the empirical discussion needs a more in-depth discussion of the contributing factor of the main components, such as
       - the fact that Q, R, P are diagonal matrices (and the assumption behind this)
       - the fact that Q, R, P are parametrized as s I, where s is a scalar.
   - In the experiments, we can see that AXBN is only marginally better than XBN in many cases, which is expected as sometimes being adaptive is not necessary, however, it seems important to discuss when and where A of AXBN is not necessary (only marginally improving)
   - Aren't there also cases when AXBN, XBN, and XBM all perform similarly? e.g., when simple datasets when the separation is clear, and not affected much by parameters' update, but the discussion does not include such cases.

**Strengths And Weaknesses:**

Strengths
   - The idea of the paper is every intuitive and needed.
   - The paper is written fairly clearly, the background of XBM is very helpful.

Weakness
   - The empirical discussion seems quite limited, despite the high final performances (see below).
   - (Minor) the paper assumes the background of XBN, where first and second-order statistics alignment is sufficient.
       - the authors state this limitation clearly, but I don't think it's crucial as long as the background is stated clearly.

---

> ### Author Response · Authors · 2023-01-25
> **Response to e7XY**
>
> We thank the reviewer for recognizing that our approach is intuitive and the writing is clear. We address the specific comments below and the manuscript is updated to incorporate the suggestions wherever applicable.
>
> ### Assumptions in the AXBN approach
>
> * As discussed in Sec. 3.2.1 and in the limitations section, we do not employ the full Kalman filter formulation but use a simplified version by making two assumptions: 1) independence among dimensions (for computational benefits), and 2) the noise variance is isometric (to reduce the number of hyperparameters). These assumptions are made to simplify the formulation and reduce computation. These are, perhaps, the most basic and general assumptions that can be made — we make no claim that they are optimal for specific tasks. Even so, the simplified version lead to (slight) performance improvement across many experiments.
> * We recognize the reviewer’s concern that the effect of these assumptions is not clear. Nevertheless, as our non-adaptive version (XBN) is already significantly outperforming the original XBM approach, and the AXBN approach is only marginally better than XBN, we believe any deep-dive investigation of AXBN may not add much value to this paper. We believe such investigations can be left for future work.
>
> ### Need for AXBN
>
> * As discussed in Sec. 5.3, the AXBN approach is useful when the sampling noise due to minibatches is high. There are two factors that affect the sampling noise: 1) minibatch size, and 2) the stability of the embedding network. The adaptive method may not be necessary when one has a stable embedding network (a network that shows slow feature-drift) and/or the minibatch size is large. We have added this discussion in Appendix E.
>
> ### Need for cross batch memory
>
> * As the reviewer mentioned, AXBN, XBN and XBM all would perform similarly in cases where the cross batch memory is not needed. As one may extrapolate from Fig. 1 top-row that larger batch sizes lead to smaller gap between the methods (see the response to reviewer o42X) and when the batch size is large enough, cross batch memory is not required. We have mentioned this in Appendix F.

---

### Decision · Action_Editors · 2023-03-02

**Recommendation:** Reject

**Comment:**

This paper aims to study the batch normalization technique for metric learning with ranking loss. This paper pointed out that the method with the ranking loss highly depends on the minibatch size. The proposed approach is based on the scheme of XBN, accumulating the embeddings across minibatch, and aims to solve the representational drift issue between the accumulated embeddings and the feature embeddings at the current training iteration as the learnable parameters are being updated. The proposed approach, called adaptive cross batch normalization, adapts the stored embeddings to match the first and second moments of the current embeddings at each training iteration. Experiments on image retrieval datasets justified the effectiveness of the proposed approach.

We appreciate the efforts from authors in their detailed rebuttal. Given the simplicity of the proposed approach and the strong empirical evidence, I personally love this idea. Although some reviewers challenge its simplicity, we should firmly follow that the novelty or significance of methods should not the selection criteria of TMLR (https://jmlr.org/tmlr/acceptance-criteria.html). However, **three qualified reviewers still tend to reject this work after reading rebuttal, while one qualified reviewer keeps positive**. Therefore, the current version still has some problems. For example, (1) the authors didn't adequately address the main concerns on the lack of clarifications of the design choices of several assumptions of the method. (2) the current manuscript even has opposite evidence on the research hypothesis that feature drifting issue is crucial because it may even change ranks and make ranking loss produce wrong signal. Evidence like the number of samples having wrong rank and relationship between rank-changing/feature-drift and final performance were requested by the reviewer to verify the hypothesis of the paper but the authors only point the reviewer to Figure 3 in the appendix, where the other compared variant XBM* has smaller drift but lower performance, which disagrees with the research hypothesis. (3) the need for the Kalman filter is not fully motivated, since no realistic low batch size setting has been provided. Having said that, the authors already provide evidence (in artificially constrained training settings) that low-batch sizes would be better handled through the Kalman filter.

Therefore, we cannot accept this work this time, but the authors are highly encouraged to **resubmit after a major and significant revision**.

**Audience:**

Yes

**Claims And Evidence:**

According to qualified reviewers, the current version still has some problems. For example, (1) the authors didn't adequately address the main concerns on the lack of clarifications of the design choices of several assumptions of the method. (2) the current manuscript even has opposite evidence on the research hypothesis that feature drifting issue is crucial because it may even change ranks and make ranking loss produce wrong signal. Evidence like the number of samples having wrong rank and relationship between rank-changing/feature-drift and final performance were requested by the reviewer to verify the hypothesis of the paper but the authors only point the reviewer to Figure 3 in the appendix, where the other compared variant XBM* has smaller drift but lower performance, which disagrees with the research hypothesis. (3) the need for the Kalman filter is not fully motivated, since no realistic low batch size setting has been provided. Having said that, the authors already provide evidence (in artificially constrained training settings) that low-batch sizes would be better handled through the Kalman filter. Therefore, the authors are encouraged to resubmit after a major revision.